# Therapeutic Implications of Menin Inhibitors in the Treatment of Acute Leukemia: A Critical Review

**DOI:** 10.3390/diseases13070227

**Published:** 2025-07-19

**Authors:** Martina Canichella, Cristina Papayannidis, Carla Mazzone, Paolo de Fabritiis

**Affiliations:** 1Hematology, St. Eugenio Hospital, ASL Roma2, 00144 Rome, Italy; carla.mazzone@aslroma2.it (C.M.); paolo.defabritiis@aslroma2.it (P.d.F.); 2IRCCS Azienda Ospedaliero-Universitaria di Bologna, Istituto di Ematologia L. e A. Seràgnoli, 40138 Bologna, Italy; cristina.papayannidis@unibo.it; 3Department of Biomedicina e Prevenzione, Tor Vergata University, 00133 Rome, Italy

**Keywords:** acute myeloid leukemia (AML), acute lymphoblastic leukemia, *KMT2A* rearrangement (*KMT2A*r), *NPM1* mutation (*NPM1*m), revumenib, ziftomenib, bleximenib, enzomenib, allogeneic hematopoietic stem cell transplantation (allo-HSCT)

## Abstract

Menin inhibitors are a class of targeted agents that exemplify how a deeper understanding of leukemia pathogenesis can unify seemingly distinct genetic acute leukemia subgroups under a common therapeutic strategy. In particular, acute leukemia with *NPM1* mutations *(NPM1*m) and *KMT2A* rearrangements (*KMT2A*r) represent the primary targets of this emerging drug class. Acute myeloid leukemia (AML) with *NPM1*m—which accounts for approximately 30% of AML cases and AML or acute lymphoblastic leukemia (ALL) with *KMT2A*r—and is present in 5–10% of cases, shares a common pathogenetic mechanism: the aberrant activation of the *MEIS1–HOXA* axis. These leukemic subsets are associated with poor prognosis, particularly in the relapsed/refractory (R/R) setting. For *KMT2A*r AML, the prognosis is especially dismal, with a median overall survival (OS) of 2.4 months and a complete remission (CR) rate of only 5%. In *NPM1*m AML, intensive chemotherapy achieves remission in approximately 80% of cases, but relapse remains a major challenge, occurring in nearly 50% of patients. Relapsed *NPM1*m AML is linked to a poor prognosis, with a median OS of 6.1 months (12-month OS: 30%) and a median relapse-free survival (RFS) of 5.5 months (12-month RFS: 34%). Menin inhibitors directly target the leukemogenic transcriptional program driven by *HOX* and *MEIS1*, disrupting oncogenic signaling and offering a promising therapeutic approach for these high-risk patients. This class of agents has rapidly progressed through clinical development, showing promising antileukemic activity in both treatment-naïve and R/R AML. Currently, six menin inhibitors are in clinical evaluation as monotherapy or in combination regimens: revumenib, ziftomenib, bleximenib (previously JNJ-75276617), enzomenib (previously DSP-5336), DS-1594, and BMF-219. In this review, we critically analyze the clinical development and therapeutic potential of the four most extensively studied menin inhibitors—revumenib, ziftomenib, bleximenib, and enzomenib. We discuss their efficacy, safety profiles, and potential roles within the current treatment algorithm. The continued clinical evaluation of menin inhibitors may redefine treatment paradigms for *NPM1*m and *KMT2A*r AML and other acute leukemia with the aberrant *MEIS1-HOXA* axis, offering new hope for patients with limited therapeutic options.

## 1. Introduction

Over the past decade, the therapeutic landscape of acute myeloid leukemia (AML) has evolved significantly with the introduction of targeted agents that have improved outcomes in both the frontline and relapsed/refractory (R/R) settings. The combination of venetoclax with hypomethylating agents, and the advent of *FLT3* and *IDH1/IDH2* inhibitors, has markedly improved overall survival (OS) in selected AML subgroups [1,2]. However, prognosis remains poor for patients with genetic alterations, such as relapsed *NPM1* mutations (*NPM1*m) and *KMT2A* rearrangements (*KMT2A*r), with median OS around six months [3,4,5,6]. Menin inhibitors have emerged as a promising strategy for these genetically defined AML subsets, targeting the oncogenic dependency on the menin–*KMT2A* interaction [7]. By disrupting this interaction, they modulate leukemic transcriptional programs and induce blast differentiation. In recent years, six menin inhibitors have progressed into clinical evaluation, demonstrating robust pre-clinical rationales (such as menin–*KMT2A/NPM1* axis disruption leading to leukemic differentiation) and clinically meaningful activity with manageable safety profiles. In the R/R AML setting, these agents have induced complete remission (CR), or a CR with partial hematologic recovery (CRh) in approximately 20–26% of heavily pre-treated patients, offering a novel targeted approach for a population with otherwise poor therapeutic prospects.

Early-phase results in the R/R setting have also supported their use in frontline therapy. In combination with standard regimens such as 3 + 7 or venetoclax-azacitidine (VEN-AZA), they have demonstrated efficacy and tolerability. This review analyzes menin inhibitors, focusing on revumenib, ziftomenib, bleximenib, and enzomenib—the four compounds with the most substantial clinical results to date. Notably, revumenib has emerged as the first-in-class menin inhibitor to obtain formal regulatory approval: the FDA granted approval on 15 November 2024 for adult and pediatric patients (≥1 year) with R/R acute leukemia carrying a *KMT2A* translocation based on the AUGMENT-101 Phase I/II trial. In this review for each menin inhibitor, we examine pharmacologic characteristics, clinical efficacy, safety profile, and their potential to redefine the treatment paradigm for *NPM1*m and *KMT2A*r AML. In addition, for each inhibitor, we clearly distinguished between clinical studies in which it was used as a single agent and those in which it was employed in combination regimens. The latter approach appears to be the most promising, as will be further discussed in the sections below.

## 2. Menin Protein: From Physiological Role to Leukemogenesis

Menin is a scaffold nuclear protein encoded by the *MEN1* gene (chromosome 11q13), and it is classically associated with hereditary *MEN1* syndrome due to its tumor suppressor role [8,9]. However, beyond this function, menin interacts with multiple proteins and transcription factors, influencing gene expression through DNA binding. This role is particularly relevant in the pathogenesis of *NPM1*m and *KMT2A*r AML, where menin acts as an oncogenic cofactor. The interaction between menin and *KMT2A* fusion proteins promotes leukemogenesis via upregulation of homeobox (*HOX*) genes and their cofactor *MEIS1* (the *HOX/MEIS1* complex). *KMT2A* (previously *MLL*), located at 11q23, encodes a transcriptional regulator of genes crucial for hematopoiesis (e.g., *HOXA, HOXB, MEIS1, PBX3, MEF2C,* and *CDK6*). Rearrangements involving *KMT2A* occur in ~10% of acute leukemias, with high prevalence in infant ALL (>70%) and involvement of over 100 fusion partners [10,11]. These fusions alter chromatin dynamics and drive leukemic transformation. In *KMT2A*r leukemias, the menin–*KMT2A* interaction is essential for nuclear translocation and transcriptional activation of leukemogenic programs. For this reason, *KMT2A*r AML is classified as high-risk by ELN 2022 guidelines, with poor response to chemotherapy and recommendation for upfront allo-HSCT [12,13,14]. Similarly, *NPM1*m AML—found in ~25–30% of adults and 10% of children—is dependent on *HOX/MEIS1* hyperactivation [1]. *NPM1*m is usually standard-risk unless adverse co-mutations (e.g., *FLT3-ITD*) are present [14]. Despite a good initial response, there is a ~50% chance of relapse. Mutations—commonly insertions in exon 12—cause cytoplasmic delocalization of *NPM1*, disrupting its nucleolar function. Transcriptomic similarities between *NPM1*m and *KMT2A*r AML support the rationale for targeting the *KMT2A–*menin axis in both subtypes. Figure 1 summarizes these mechanisms.

### 2.1. Revumenib

Revumenib (formerly SNDX-5613) was the first menin inhibitor to demonstrate promising efficacy in preclinical studies and xenograft models by suppressing blast proliferation and promoting cellular differentiation. On 15 November 2024, revumenib received the FDA approval for R/R acute leukemia (AL) with *KMT2A*r in adult and pediatric patients 1 year and older [15]. The clinical results of main trials experimenting with revumenib in R/R or front-line *NPM1*m and *KMT2A*r AL approval are discussed below and summarized in Table 1.

#### 2.1.1. Revumenib Single Agent

Revumenib was evaluated as a monotherapy in the AUGMENT-101 trial (NCT04065399), a phase 1/2 open-label study assessing safety and efficacy in adult and pediatric patients with R/R acute leukemia harboring *NPM1*m or *KMT2A*r [16]. Phase 1 enrolled patients in two arms: Arm A included those not receiving strong CYP3A4 inhibitors, while Arm B included those receiving them. Revumenib was administered orally every 12 h on Days 1–28 of a 28-day cycle. Arm A tested four dose levels (113 mg, 226 mg, 276 mg, and 339 mg), whereas Arm B tested three (113 mg, 226 mg, and 276 mg). The primary objectives of Phase 1 were to assess the safety, maximum tolerated dose (MTD), recommended Phase 2 dose (RP2D), and pharmacokinetics (PK). A total of 68 patients were enrolled in Phase 1, with a median age of 42.5 years; of them, 60 were adults and 8 were pediatric patients. The median number of prior treatment lines was four, with 46% of patients relapsing after allogeneic hematopoietic stem cell tranplanstation (allo-HSCT). The overall response rate (ORR) was 53%, with complete remission (CR) or CR with partial hematologic recovery (CRh) observed in 30% of cases. The median time to CR/CRh was 1.9 months (range, 0.9–4.9). Among CR/CRh responders, measurable residual disease (MRD) negativity was achieved in 78%. The median OS for the entire cohort was seven months, and 12 patients proceeded to allo-HSCT. The median duration of response (DoR) was 9.1 months. RNA sequencing analysis demonstrated downregulation of leukemogenic genes, including *HOXA/MEIS1*, and upregulation of differentiation-related genes. Safety assessments identified QTc prolongation as the only dose-limiting toxicity (DLT), occurring at any grade in 53% of patients, with grade 3 or 4 events in 13%. All QTc prolongations were reversible, and no ventricular arrhythmias were reported. Differentiation syndrome (DS) was observed in 16% of patients, all cases being grade 2.

In Phase 2, patients with *KMT2A*r leukemia received 163 mg (or 95 mg/m^2^ if <40 kg) twice daily plus a strong CYP3A4 inhibitor [20]. Interim results (n = 57) showed CR/CRh in 22.8%, with a median DoR of 6.4 months. The CRc rate was 43.9%, ORR 63.2%, and MRD negativity was 68.2% in the evaluable patients. Allo-HSCT was performed in 38.9%, with half resuming revumenib post-transplant. Updated Phase 2 data (n = 116) showed CR/CRh in 23% (DoR: 6.4 months); composite CR (CRc: CR + CRh + CR with incomplete platelet recovery + CR with incomplete count recovery) in 42%; ORR 64%; and MRD negativity in 61% of CR/CRh and 58% of CRc responders [21]. Among the ORR patients, 34% underwent allo-HSCT, and nine restarted revumenib post-transplant. Grade ≥ 3 TEAEs occurred in 91% and TRAEs in 54%, with febrile neutropenia being the most common (39%). DS was reported in 15% (one grade 4, no grade 5 events) [17,22,23].

#### 2.1.2. Revumenib in Combination with Other Agents

The subsequent AUGMENT-102 trial (NCT05326516) investigated revumenib in combination with fludarabine and cytarabine (FLA) in 27 patients with R/R AML harboring *NPM1*m, *KMT2A*r, or *NUP98* rearrangements [17]. CRc was achieved in 56% and 50% of patients at the 113 mg and 163 mg doses, respectively. MRD negativity was observed in 71% of evaluable patients, and seven patients proceeded to allo-HSCT. Severe (grade ≥ 3) AEs occurring in >40% included thrombocytopenia (63%), anemia (56%), and febrile neutropenia (48%). Cytopenias were less frequent at 163 mg compared to 113 mg, correlating with a faster remission induction. No cases of DS were reported. These findings supported the selection of revumenib at 163 mg q12h (or 95 mg/m^2^ for patients < 40 kg) as the RP2D in combination with FLA and a strong CYP3A4 inhibitor, aligning with the monotherapy dose under FDA review.

The BEAT AML trial evaluated revumenib with VEN-AZA in newly diagnosed AML patients aged >60 years with *KMT2Ar* or *NPM1m* [19]. During dose escalation, revumenib was administered at 113 mg or 163 mg q12h with a strong CYP3A4 inhibitor alongside venetoclax and azacitidine. As of 1 May 2024, 26 patients had been enrolled. Among efficacy-evaluable patients, the CRc rate was 96% (23/24), with 92% achieving MRD negativity. Three patients proceeded to allo-HSCT. The 12-month OS estimate in the first cohort was 100%. Revumenib was well-tolerated at both doses. DS occurred in 15% (one grade 3 case). QTc prolongation was observed in 46% (three grade 3 cases). All cases resolved without complications or dose reductions. The safety profile was consistent with prior VEN-AZA studies, with no additional safety concerns arising from the combination. These findings support further investigation of revumenib in frontline AML therapy.

#### 2.1.3. Considerations on Revumenib

Revumenib was the first clinically tested menin inhibitor, offering valuable insights into its pharmacokinetic and pharmacodynamic properties, as well as the management of adverse events associated with this drug class. Its mechanism of action appears to induce blast differentiation. Notably, patients with *KMT2A*r leukemia who achieved CR with MRD negativity, still exhibited detectable fusion transcripts, a dynamic response resembling that observed in acute promyelocytic leukemia (APL) and other targeted therapies, after one treatment cycle. From a pharmacokinetic perspective, early studies demonstrated significant variability in drug metabolism among patients receiving concomitant strong CYP3A4 inhibitors, such as posaconazole or voriconazole. Revumenib has also refined the management of two primary menin inhibitor-related TRAEs: DS and QTc prolongation. DS required prompt corticosteroid administration, along with hydroxyurea, when white blood cell counts exceeded 25 × 10^9^/L, and this was without necessitating treatment discontinuation. QTc prolongation management included electrolyte correction, withholding revumenib for QTc ≥ 481 ms, and dose reduction if the prolongation persisted beyond two weeks. The results of the AUGMENT-101 trial are particularly encouraging given the dismal prognosis typically associated with heavily pretreated KMT2Ar-AL, where median OS rarely exceeds 2.4 months and CR rates are below 5%. Beyond demonstrating clinical activity, the trial highlighted three key therapeutic implications of revumenib: (1) over half of the patients achieving CR or CRh were MRD negative, a finding associated with improved survival outcomes—especially when followed by allo-HSCT; (2) revumenib may serve as an effective bridge to transplantation in eligible patients, thereby prolonging survival; and (3) its emerging role in post-transplant maintenance therapy, where the aim is to prevent relapse in this high-risk molecular subgroup. With regard to the subsequent AUGMENT-102 trial, which investigated revumenib in combination with FLA in the R/R setting, an interesting issue is that it included patients with *NUP98*-rearranged AML, highlighting the pivotal and transversal role of the *HOXA-MEIS1* axis in leukemogenesis as a key oncogenic driver across multiple AML subtypes. *NUP98*, located on chromosome 11p15, interacts with the histone methyltransferase *NSD1*, and preclinical studies have demonstrated that these leukemias are dependent on *KMT2A* activity, which can be disrupted through menin inhibition [24]. Revumenib in combination with FLA has demonstrated substantial efficacy, particularly at the 163 mg q12h dose, with an excellent safety profile and a notable absence of DS, suggesting that combination therapy may mitigate drug-related adverse effects. Furthermore, emerging data from the BEAT AML study indicate that frontline revumenib in combination with the VEN-AZA regimen represents a highly promising approach, achieving the most favorable responses in older patients (>60 years) who may be unfit for intensive chemotherapy. The integration of revumenib into the treatment paradigm for *NPM1*m and *KMT2Ar* AML has critical implications for allo-HSCT eligibility and post-transplant maintenance strategies. Several additional combination studies are ongoing, with results anticipated in the near future. Notably, a particularly compelling area of investigation is the use of revumenib in the MRD setting in combination with venetoclax, which is currently being explored in the ongoing NCT06284486 trial.

### 2.2. Ziftomenib

Ziftomenib (KO-539) is an orally selective menin inhibitor that has shown the ability to promote differentiation and suppress the proliferation of leukemic cells in preclinical models. The results of different KOMET trials that investigated ziftomenib are illustrated below and summarized in Table 2.

#### 2.2.1. Ziftomenib Single Agent

KOMET-001 (NCT04067336) is a multicenter phase 1/2 trial evaluating ziftomenib in adults with R/R AML harboring *KMT2A*r or *NPM1*m. The primary aim was to establish the RP2D based on safety, PK, pharmacodynamics, and preliminary efficacy [25]. The primary aim of this study was to establish the RP2D based on safety, pharmacokinetics, pharmacodynamics, and preliminary efficacy. Recently published findings include data from Phase 1a (dose-escalation) and Phase 1b (dose-validation) studies. During Phase 1a, various doses of ziftomenib (50–1000 mg daily in 28-day cycles) were assessed across all AML molecular subtypes (n = 30). Phase 1b subsequently enrolled 53 patients with *NPM1*m or *KMT2Ar*, who were randomized (1:1) into two cohorts evaluating two Phase 2 dose candidates (200 mg and 600 mg). In Phase 1a, two DLT were identified: grade 3 pneumonitis at 400 mg and grade 4 DS followed by grade 5 cardiac arrest at 1000 mg. Given that grade 4 DS predominantly affected patients with *KMT2Ar* and was more severe than in those with *NPM1m*, Phase 1b enrollment was restricted to *NPM1m* cases. In Phase 1b, the 200 mg dose did not result in a reduction in blast counts. At the 600 mg recommended dose, 9 (25%) of 36 patients with either *KMT2A*r or *NPM1*m achieved CR or CRh. Among patients with *NPM1*m, 7/20 (35%) patients treated at 600 mg achieved CR. Interestingly, 2 of 20 patients with *NPM1*m treated with 600 mg ziftomenib received allo-HSCT and remained in remission as of the cutoff date, where 1 received post-transplantation ziftomenib maintenance [25].

#### 2.2.2. Ziftomenib in Combination with Other Agents

In the ongoing KOMET-007 trial, ziftomenib was combined with venetoclax and azacitidine in R/R AML [26]. Among 54 patients (26 *NPM1*m, 28 *KMT2A*r), TEAEs occurred in ≥20%, with no QTc prolongation. DS was reported in 8% (all grade 2–3 and manageable). In NPM1m, ORR was 68% and CRc 50%; in KMT2Ar, ~33% responded. Responses were observed even after prior venetoclax exposure. In this subgroup, ORR and CRc were 50% and 36%, respectively [26]. Ziftomenib was also tested with 7+3 chemotherapy in newly diagnosed high-risk AML [27]. Among 54 patients (24 *NPM1*m, 27 *KMT2A*r), no DLTs or QTc prolongation were reported. Median CR and OS were not reached in either group. In *NPM1*m, 5 underwent allo-HSCT (2 on maintenance)—all remained alive. In *KMT2A*r, 10 received allo-HSCT (5 on maintenance), and 96% (26/27) were alive at cutoff [27].

KOMET-008 is an ongoing Phase 1 study assessing ziftomenib in combination with gilteritinib, FLAG-IDA, or LDAC in R/R AML with NPM1m or KMT2Ar. One cohort includes *NPM1*m/*FLT3*-mutated patients treated with ziftomenib plus gilteritinib [28].

#### 2.2.3. Considerations on Ziftomenib

Ziftomenib has demonstrated to be a promising and effective drug with an acceptable safety profile. Its introduction in the treatment of heavily pretreated patients offers them the opportunity to achieve deep remission, including MRD negativity. Within the therapeutic algorithm of these patients, ziftomenib emerges as a bridge to allo-SCT and a valuable maintenance therapy option in the post-transplant setting. As previously demonstrated with revumenib, ziftomenib’s greatest efficacy derives from its synergistic effect when combined with other targeted agents or chemotherapy regimens. Notably, the combination of ziftomenib with the VEN-AZA regimen in the setting of R/R patients has shown a restoration of response, even in those previously exposed to VEN. VEN belongs to the class of BCL-2 inhibitors, which bind to the *BH3* domain of *BCL-2*, thereby promoting apoptosis. The VEN-AZA combination is currently approved as a frontline therapy for unfit patients and remains a valuable option in the relapsed setting. Nevertheless, resistance to venetoclax poses a significant therapeutic challenge. In this context, the potential role of ziftomenib in resensitizing leukemic cells to BCL-2 inhibition represents a promising therapeutic advance. Although specific data in venetoclax-resistant models are still lacking, the underlying rationale is compelling: by promoting leukemic cell differentiation, ziftomenib may enhance cellular susceptibility to venetoclax and help overcome established resistance mechanisms. Future studies, both at the molecular biology level and in clinical settings, will be essential to further elucidate this potential mechanism and define the role of ziftomenib in overcoming venetoclax resistance.

### 2.3. Bleximenib

Bleximenib (JNJ-75276617) is a potent, orally bioavailable, and selective inhibitor of the interaction between *KMT2A* and menin. In preclinical studies, bleximenib demonstrated the ability to reduce the expression of menin-*KMT2A* target genes, including *MEIS1*, and it increased differentiation markers in blast cells [29,30]. We report below the most important results of the main clinical trial that investigated bleximenib (Table 3).

#### 2.3.1. Bleximenib Single Agent

At EHA 2022, Jabbour E. presented preliminary results from the Phase 1/2 cAMeLot study (NCT04811560), evaluating bleximenib monotherapy in patients with R/R AML or ALL harboring *NPM1m* or *KMT2A*r [31]. A total of 58 patients (median age 63; range 19–83) were enrolled and treated in a 28-day dose-escalation design. The median number of prior therapies was two (range 1–7), and 17% had received prior allo-HSCT. TRAEs occurred in 52% (29% grade ≥ 3) and DS in 14% (5% grade ≥ 3). Pharmacodynamic data showed downregulation of menin-*KMT2A* targets. No RP2D was established; dose escalation remains ongoing. These initial findings supported bleximenib’s activity and safety in R/R AML [31]. At ASH 2024, Searle E. presented updated data to define the RP2D from Phase 1 of cAMeLot [32]. By July 2024, 121 patients had been enrolled (median two prior therapies; 25% post-allo-HSCT). Three dose levels were tested: 45 mg BID (n = 15), 90/100 mg BID (n = 27), and 150 mg BID (n = 28). TRAEs occurred in 58% overall; DS (13%) was the most common, with 7% experiencing grade ≥3 DS (2 fatal). Dose interruptions/reductions were most frequent at 150 mg BID (25% and 14%) compared to lower doses (7% and 7%). The ORR was 50% at both 90/100 mg BID and 150 mg BID and 39% at 45 mg BID. At 90/100 mg BID, the median time to first response was 30 days (range: 27–85), with a median DoR of 6.4 months. Responses were comparable in *NPM1*m and *KMT2A*r. The RP2D was defined as 100 mg BID after a 50 mg BID step-up dose, balancing efficacy and tolerability. These results support ongoing trials of bleximenib in combination regimens [32].

#### 2.3.2. Bleximenib in Combination with Other Agents

In a Phase 1b study presented by Wei et al. at EHA 2024, bleximenib was administered at doses of ≥15 mg BID in combination with VEN-AZA, starting on Day 4 continuously [34]. A total of 45 patients received the triplet combination (bleximenib + VEN + AZA), with a median age of 60 years (range: 20–82). Safety analysis showed that TRAEs of any grade occurred in 22% (10/45) of patients, with the most frequent being grade ≤ 2 gastrointestinal (GI) events (7%) and one case of grade 3 hyperkalemia (2%). No cases of DS, QTc prolongation, tumor lysis syndrome (TLS), or DLTs were reported. In the efficacy dataset for patients receiving doses of ≥50 mg BID, the ORR was 86% (18/21), with a CRc rate of 48% (10/21) and a CR/CRh rate of 24% (5/21). ORR was consistent across genetic subgroups. Among patients with prior VEN exposure, ORR was 82% (9/11), CRc was 36% (4/11), and CR/CRh was 18% (2/11). The median time to first response in the efficacy dataset was 23 days (range: 14–59), and the median time to cCR was 52 days (range: 14–100). Promising results were also observed with the combination of bleximenib and standard induction chemotherapy (3 + 7 regimen) in newly diagnosed AML harboring *NPM1*m or *KMT2A*r. In this Phase 1b, multicenter, and dose-finding study, bleximenib was administered with cytarabine (200 mg/m^2^/day) and daunorubicin (60 mg/m^2^/day IV) or idarubicin (12 mg/m^2^/day IV) [33]. Patients who achieved CR received consolidation therapy with up to four cycles of intermediate-dose cytarabine plus bleximenib. Those who did not proceed to allo-HSCT were eligible for bleximenib maintenance therapy for up to 12 months. A total of 22 patients were analyzed for safety, with 95% (21/22) experiencing at least one TRAE. The most frequent TRAEs were diarrhea (17/22; 77%) and thrombocytopenia (15/22; 68%). Grade ≥3 TRAEs attributed to bleximenib monotherapy occurred in 14% (3/22) of patients, including thrombocytopenia (2/22; 9%). No DS or DLTs were observed. The efficacy analysis showed an ORR of 93%, consistent across genetic subtypes, with a median time to CR of 30 days (range: 22–41). Among the five patients who completed consolidation therapy, all proceeded to allo-HSCT.

#### 2.3.3. Considerations on Bleximenib

Based on the results of the clinical studies discussed above, bleximenib has demonstrated clinically meaningful activity in terms of both safety and efficacy. Notably, its use in the R/R AML setting has yielded results comparable to those of other menin inhibitors, despite being evaluated in a heavily pretreated population. Furthermore, bleximenib-based combination therapies appear highly promising, with no evidence of added toxicity. The triplet regimen of bleximenib, VEN, and AZA in R/R NPM1m and KMT2Ar and patients has been associated with generally manageable adverse events and encouraging responses, including in patients previously exposed to VEN. This suggests a potential re-sensitization effect, as previously observed with ziftomenib [29]. A key distinguishing feature of bleximenib compared to other menin inhibitors is its ability to overcome specific resistance mutations, particularly MEN1^M327I and MEN1^T349M. These mutations disrupt menin inhibitor binding and represent a well-characterized resistance mechanism within this drug class. Preclinical data indicate that bleximenib retains activity against these variants, potentially providing an advantage over other menin inhibitors, particularly in patients who develop secondary resistance. Most cases of resistance to menin inhibitors have been attributed to MEN1 mutations, particularly MEN1^M327I and MEN1^T349M, which result in amino acid substitutions within the inhibitor-binding pocket (specifically affecting Trp346, a critical residue for drug binding). Interestingly, JNJ-75276617 appears to exhibit a distinct binding mode, relying on interactions within the flexible tail region of the compound and depending on Asp290. This alternative binding mechanism allows the compound to retain antiproliferative activity, even in the presence of common MEN1 resistance mutations. In conclusion, bleximenib emerges as a promising therapeutic option with significant anti-leukemic activity, a manageable safety profile, and the potential to overcome key resistance mechanisms. However, it is important to note that the follow-up duration is short, and the activity in patients previously treated with menin inhibitors remains uncertain.

### 2.4. Enzomenib

Enzomenib (DSP-5336) is a menin inhibitor that has demonstrated potent anti-leukemic activity in preclinical studies by downregulating *HOXA9* and *MEIS1*, key genes involved in leukemogenesis [35]. Table 4 details the main clinical studies with enzomenib. At the 2024 ASH meeting, Zeidner et al. presented preliminary results from a Phase 1/2 trial evaluating enzomenib monotherapy in patients with R/R *NPM1*m, *KMT2A*r, and other *HOXA9/MEIS1*-driven leukemias [36]. Dose escalation and optimization were conducted in two parallel arms: Arm A (without strong CYP3A4-inhibiting azole antifungals) and Arm B (with strong CYP3A4 inhibitors). The primary endpoints were safety and tolerability in Phase 1 and efficacy in Phase 2. A total of 84 patients were enrolled, including 31 in Arm A and 50 in Arm B, with a median age of 60 years (range: 20–89). The majority (93.8%) had AML. The median number of prior lines of therapy was 3, including 23 patients (28.4%) who had undergone allo-HSCT, 63 patients (77.8%) who had received prior venetoclax, and 6 patients (7.4%) who had been treated with a prior menin inhibitor. During Phase 1, enzomenib was escalated from 40 mg BID to 300 mg BID without DLTs. QTc prolongation of ≥grade 3 related to enzomenib was not observed, while DS occurred in 9 patients (11.1%), though no fatal cases or treatment-related deaths were reported. Among the 35 patients who were menin inhibitor-naïve and received active doses of enzomenib (≥140 mg BID in Arm A or Arm B), responses were notable. In the 22 patients with *KMT2Ar* AML, the ORR was 59.1% (13/22), with CR + CRh achieved in 22.7% (5/22). Similarly, among 13 patients with *NPM1m* AML, the ORR was 53.8% (7/13), with CR + CRh achieved in 23.1% (3/13). Interestingly, no significant drug accumulation was observed, suggesting that azole antifungals do not substantially impact enzomenib exposure. While the RP2D for monotherapy has not yet been established, preliminary efficacy signals suggest that doses between 140 mg and 300 mg BID may be within the optimal therapeutic range.

#### Consideration of Enzomenib

Enzomenib (DSP-5336) is a distinct menin inhibitor, characterized by a favorable chemical properties and pharmacological profile, which sets it apart from other agents in this class. Enzomenib demonstrates dose-dependent pharmacokinetics with a plasma half-life of ~2–10 h, rapid Tmax (~2 h), and negligible accumulation with repeated BID dosing. Pharmacodynamic assessments in KMT2A-rearranged or NPM1-mutated AML patients has revealed swift downregulation of the stemness markers HOXA9, MEIS1, and PBX3, as well as the upregulation of differentiation marker CD11b. These PK/PD characteristics—predictable exposure, minimal drug–azole interaction, and robust target engagement—support a favorable therapeutic index for enzomenib monotherapy. Importantly, enzomenib has demonstrated significant anti-leukemic activity, inducing promising clinical responses supported by molecular studies that confirm the activation of leukemic differentiation pathways. Moreover, preliminary data suggest that enzomenib may have a more favorable safety profile compared to other menin inhibitors, which could be a key differentiating factor in clinical practice. However, its overall efficacy and tolerability require further validation. A critical area of ongoing investigation is its activity in patients previously exposed to other menin inhibitors, as cross-resistance remains a potential concern. Upcoming clinical and translational data will be crucial in clarifying enzomenib activity.

## 3. Discussion

The clinical studies on revumenib, ziftomenib, bleximenib, and enzomenib have undoubtedly shown promising data; however clinical trials on revumenib and ziftomenib have already provided encouraging preliminary results, but those evaluating bleximenib and enzomenib—especially in newly diagnosed patients—are still in the enrollment phase, with only small cohorts currently under treatment and limited follow-up data available. Taken together, all of these results suggest that menin inhibitors could represent a valid therapeutic strategy for patients with *NPM1*m or *KMT2Ar* acute leukemia, and they are expected to be incorporated into the therapeutic algorithm in the near future. From a safety perspective, all four agents have demonstrated an acceptable toxicity profile, with manageable adverse events. The most common toxicities, such as DS and cytopenias, appear to be manageable, without major off-target effects that would limit their clinical use. In terms of efficacy, these inhibitors show similar activity, and their mechanism of action is strongly supported by molecular studies. These studies confirm that menin inhibitors reshape the leukemic transcriptome, leading to the upregulation of genes responsible for differentiation and promoting a shift towards a more normal hematopoietic phenotype. The clinical application of these agents appears to be broad, as they have shown promising activity in different treatment settings. In the R/R setting, where they have been primarily developed, they have demonstrated meaningful clinical responses, even in patients heavily pretreated, including those who have undergone allo-HSCT. An additional key aspect is their role in combination therapy. In an R/R setting, the association of menin inhibitors plus the VEN-AZA regimen demonstrated significant responses induced in the patients and may also restore sensitivity to venetoclax in patients who have been previously exposed. This is particularly relevant given the widespread use of venetoclax-based regimens and the challenge of resistance development. Another crucial role of menin inhibitors is their potential use as a bridge to transplantation. More mature follow-up data with ziftomenib and revumenib suggest that these agents can induce remission and facilitate allo-HSCT, making them valuable tools in optimizing treatment strategies for high-risk patients. Additionally, their use in post-transplant maintenance is being explored as a strategy to reduce relapse risk and improve long-term outcomes. Looking ahead, the integration of menin inhibitors into frontline therapy is another area of interest. Early results suggest that these agents can be successfully combined with intensive chemotherapy without adding significant toxicity while maintaining high CR rates. However, several critical questions remain, including the optimal patient selection beyond NPM1m and KMT2Ar AML, mechanisms of primary and secondary resistance, and the long-term durability of responses, particularly for patients not proceeding to transplant. Overall, menin inhibitors represent a promising approach in the treatment of genetically defined acute leukemias. Their efficacy, manageable safety profile, and versatility across different treatment settings highlight their potential to reshape the therapeutic strategy of specific AML subsets. As clinical trials continue to mature, these agents may become a fundamental component of the treatment algorithm, offering new hope for patients with limited therapeutic options. In addition to the aforementioned ongoing clinical trials, there are other menin inhibitors under investigation, such as DS-1594 and BFM-219. DS-1594 has demonstrated high antileukemic activity in preclinical studies and is being clinically evaluated in R/R NPM1 or KMT2A AML and ALL in combination with chemotherapy—mini-HCVD, VEN, or AZA. The results are not yet disclosed (NCT04752163) [37]. BMF-219 is an irreversible inhibitor of menin-KMT2A and is able to inhibit the activation of MYC pathways. For this reason, the ongoing Phase 1–2 clinical trial (NCT05153330) enrolled patients with NPM1m, KMT2Ar and multiple myeloma (MM), diffuse large B-cell lymphoma (DLBCL), and chronic lymphoid leukemia (CLL) [38]. Preliminary results show a good safety profile. More definitive data on efficacy are awaited [39]. The resistance of menin inhibitors represents an important challenge that risks nullifying long-term response. Molecular studies utilizing advanced technologies, such as CRISPR-Cas9 base editing, have demonstrated the emergence of somatic hotspot mutations, which prevent the binding of these inhibitors to the menin complex. Furthermore, resistance to menin inhibitors may also arise from the activation of alternative gene expression pathways independent of the MEIS–HOXA axis [40]. Notably, as previously mentioned, certain menin inhibitors, such as bleximenib, may overcome the effects of these resistance-conferring mutations. Different strategies have been applied to prevent resistance, including combination therapy with synergistic agents or novel drugs. A promising novel approach entails the co-administration of menin inhibitors with *IKAROS* degraders. *IKAROS* has been identified as a critical transcription factor regulating the MEIS/HOXA complex [41]. Mezigdomide, a next-generation IKAROS degrader belonging to the cereblon E3 ubiquitin ligase modulator (CELMoD) class, has been evaluated in preclinical models in combination with menin [42].

## 4. Conclusions

Current clinical trial results on the use of menin inhibitors, although still preliminary, have demonstrated significant anti-leukemic activity and a favorable safety profile, with manageable AEs. However, these efficacy and safety data require further validation through longer follow-up and broader clinical application. Contextually, additional therapeutic strategies must be explored to prevent or overcome the emergence of resistance-conferring mutations. Lastly, the clinical application of menin inhibitors is gradually expanding to include other acute leukemia subtypes characterized by MEIS1-HOXA axis activation, such as specific AML subgroups with particularly poor prognosis like NUP98-r, DEK-NUP21, and UBTF-TD. These genetically defined AML subgroups are associated with particularly adverse outcomes and share a common pathogenic axis driven by aberrant activation of the HOX/MEIS1 transcriptional program. These genetic subsets currently lack effective therapeutic options capable of inducing deep and durable complete responses, for which menin inhibitors could represent a promising treatment strategy. Preliminary clinical data, including responses observed with revumenib and ziftomenib in early-phase trials (e.g., AUGMENT-101), support the extension of this therapeutic approach beyond KMT2A-rearranged and NPM1-mutated AML.

## Figures and Tables

**Figure 1 diseases-13-00227-f001:**
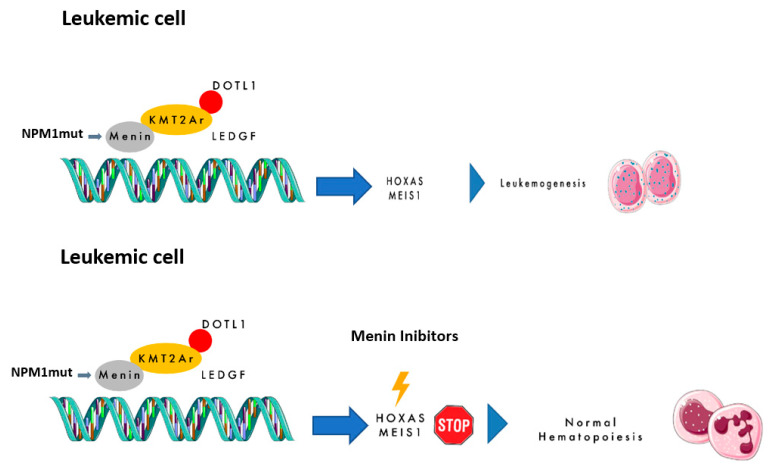
The mechanism of action of menin inhibitors.

**Table 1 diseases-13-00227-t001:** The main clinical trials of revumenib alone and in combination therapy.

Compound	Setting	Clinical Trial (NCT)	Study Population (N)	Findings	Safety	Ref.
Revumenib	Monotherapy	AUGMENT-101 NCT04065399	R/R NPM1m and KMT2Ar acute leukemias efficacy evaluation N = 97 safety evaluation N = 116	ORR 53% CR + CRh 30% MRD negativity 78%	QTc 53% DS 16%	[16]
	Combination with FLA	AUGMENT-102 NCT05326516	R/R NPM1m, KMT2Ar or NUP98r acute leukemias N = 27	CRc 51% MRD negativity 71%	TEAEs 40%	[17]
	Combination with venetoclax and azacitidine	BEAT-AML NCT03013998	newly diagnosed AML patients aged >60 years with KMT2Ar or NPM1m N = 26	CRc 96% MRD negativity 92%	DS 15% QTc 46%	[18,19]

AL, acute leukemia; ORR, overall response rate; CR, complete remission; CRh, complete remission with partial hematologic recovery; CRh CR with partial hematologic recovery, CRi, complete remission with incomplete count recovery; KMT2A, histone-lysine n-methyltransferase 2A; NPM1, Nucleophosmin 1; DS, differentiation syndrome; FLA, fludarabine and cytarabine; and TEAEs, treatment emergent adverse events.

**Table 2 diseases-13-00227-t002:** The main clinical trials of ziftomenib alone or in combination.

Compound	Setting	Clinical Trial (NCT)	Study Population	Findings	Safety	Ref.
			(N)			
Ziftomenib	Monotherapy	KOMET-001 NCT04067336	R/R KMT2Ar and NPM1m AML	CR + CRi 9/36 (25%) at 600 mg	Pneumonia G3 at 400 mg DS G4 at 1000 mg	[25]
	Combination with VEN-AZA Combination with 3 + 7	KOMET-007 NCT05735184	Newly diagnosed or R/R NPM1m or KMT2Ar AML N = 54 (ven-aza) N = 51 (3 + 7)	VEN-AZA arm ORR 68% CRc 50% 3 + 7 arm Median CR and OS were not achieved	DS 8% TEAEs > 20% TAEA > 30%	[26,27]
	Combination with gilteritinib, FLAG-IDA, LDAC *NPM1*m AML	KOMET-008 NCT06001788	R/R NPM1-m or KMT2A-r AML	NA	NA	[28]

ORR, overall response rate; CR, complete remission; CRh, complete remission with partial hematologic recovery; CRp, complete remission with incomplete platelet recovery, CRi, complete remission with incomplete count recovery; *KMT2A*, histone-lysine n-methyltransferase 2A; *NPM1* Nucleophosmin 1; DS, differentiation syndrome; FLA, fludarabine and cytarabine; and TEAEs, treatment emergent adverse events.

**Table 3 diseases-13-00227-t003:** The main clinical trials of bleximenib alone and combination therapy.

Compound	Setting	Clinical Trial (NCT)	Study Population	Findings	Safety	Ref
			(N)			
Bleximenib	Monotherapy	cAMeLot-1	R/R AML	ORR 50% (10/20)	TRAE 58%	[31,32]
		NCT 04811560	KMT2Ar NPM1m NUP98-214 N = 58	At both 90/100 mg BID and 150 mg BID, 39% (5/13) at 45 mg BID	DS 13%	
	Combination with 3 + 7	Ale1002	Newly diagnosed AML	ORR 93%	95% (21/22) > 1 TAEA 17/22; (77%) diarrhea and 15/22; (68%) thrombocytopenia	[33]
		NCT 05453903	N = 22			
	Combination with VEN-AZA	Ale1002	R/R AML KMT2Ar NPM1m	ORR: 86%	Grade ≤ 2 GI events (7%) no DS or QTc	[34]
		NCT 05453903	N = 45 pts	CRc: 48%		

ORR is defined as complete remission (CR) + CR with incomplete hematologic recovery (CRh) + CR with incomplete platelet recovery (CRp) + CR with incomplete count recovery (CRi) + morphological leukemia-free state + partial remission. CRc: composite complete response is defined as CR + CRh + CRi + CRp.

**Table 4 diseases-13-00227-t004:** The main clinical trials of enzomenib alone and combination therapy.

Compound	Setting	Clinical Trial (NCT)	Study Population	Findings	Safety	Ref.
			(N)			
Enzomenib	monotherapy	NCT04988555	R/R AML NPM1m	CR + CRh:	No QTc prolongation	[36]
			KMT2Ar and other HOXA9/MEIS1	KMT2A 7/23 (30%)	DS 11%	
			84 pts	NPM1 8/17 (47%)		

ORR is defined as complete remission (CR) + CR with incomplete hematologic recovery (CRh) + CR with incomplete platelet recovery (CRp) + CR with incomplete count recovery (CRi) + morphological leukemia-free state + partial remission. CRc: composite complete response is defined as CR + CRh + CRi + CRp.

## Data Availability

Not applicable.

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
