# Peer review of "Therapeutic Implications of Menin Inhibitors in the Treatment of Acute Leukemia: A Critical Review"

_diseases, 2025, doi:10.3390/diseases13070227_

Round 1
Reviewer 1 Report
Comments and Suggestions for Authors
The authors summarize the implications of menin inhibitors in the treatment of acute leukemia. The review article describe the six menin inhibitors in clinical evaluation.
The article is clear and concise in the scope. My comments are below:
-Can the authors add sentences summarizing the need for combination strategy? This refer to the inhibitors in evaluation in combination regimens.
-Can the authors expand the possible activity of menin inhibitors in non-NPM1 mutant and KMT2Ar AML? The authors mentioned other subtypes but can they elaborate more on them?
-Genes need to be Italicized.
Author Response
Comment: Can the authors add sentences summarizing the need for combination strategy? This refer to the inhibitors in evaluation in combination regimens.
Response: Thank you for this valuable suggestion. We fully agree that the combination strategy represents the most appealing and potentially effective therapeutic approach. We have accordingly revised the manuscript to emphasize this concept more clearly. Specifically, we have added a sentence highlighting the rationale and relevance of combination therapy prior to the sections discussing single-agent and combined treatments separately
Comment:Can the authors expand the possible activity of menin inhibitors in non-NPM1 mutant and KMT2Ar AML? The authors mentioned other subtypes but can they elaborate more on them?
Response: Thank you for this insightful comment. The potential applicability of menin inhibitors beyond KMT2A-rearranged or NPM1-mutated AML—given their ability to target a key leukemogenic axis—is indeed a fascinating aspect. While efficacy studies in other AML subgroups are still ongoing, we have cited the most representative and mature data available. A more detailed discussion of this broader potential would, however, fall outside the scope of the present review, which aims to provide an overview of the main inhibitors currently under investigation and to highlight their possible future implications.
Comment: Genes need to be Italicized
Response: Thank you, we correct them.
Reviewer 2 Report
Comments and Suggestions for Authors
The paper addresses a very important problem related to the treatment of AML, which results from the heterogeneity of this leukemia. The potential therapeutic target discussed in this publication is menin.
The paper contains dry information on the results of clinical trials on menin inhibitors. There is no broader discussion of their mechanism of action, differences between them. What are the structural differences and how they may affect their action as inhibitors. The authors should explain all abbreviations and correct punctuation errors, also in the figure. Authors should also correct the reference list in accordance with the editorial guidelines.
Author Response
Comment: The paper contains dry information on the results of clinical trials on menin inhibitors. There is no broader discussion of their mechanism of action, differences between them. What are the structural differences and how they may affect their action as inhibitors. The authors should explain all abbreviations and correct punctuation errors, also in the figure. Authors should also correct the reference list in accordance with the editorial guidelines.
Response: Thank you for your helpful suggestions. We will carefully address the punctuation issues, abbreviation consistency, and bibliography formatting during the final editing phase. Regarding the interactions between different inhibitors, we chose not to elaborate extensively on this aspect, primarily due to word count limitations and the scope of the review. However, in line with your observation, we have included a dedicated paragraph at the beginning of the manuscript outlining the underlying biological mechanisms, to provide the necessary context for the subsequent discussion.
Reviewer 3 Report
Comments and Suggestions for Authors
This manuscript addresses an important and timely topic, due to the fact that Menin inhibitors promise to open a new field in the treatment of relapsed NPM1mut AML.
The work is well organized. No references from year 2025, though.
Figure 1: can be ambiguous. Recommending to label this image as panel B, and generate a panel A that allows readers to follow the intended description. Furthermore, write a few sentences in the legend, to let readers understand the figure. Despite being familiar with the topic, it took me to read over four times to follow the image. It may have been easier if the figure was instead divided into two panels.
The current format could have been more suitable, after editing for something like a graphical abstract, which indeed has to be understood without legend; the aim then would be to provoke someone to read the article, rather than to understand before reading.
Other: based on the email address, the spelling for the name Cristina Papyannidis is wrong and needs to be corrected to Cristina Papayannidis.
I suggest a thorough proofreading of the entire manuscript.
Examine the possibility of adding recent papers:
On the other hand, most of the papers of 2025 may have been covered by the 2024 papers that are already cited in the manuscript. This is therefore only tentative, but in remote aspects it might help.
Example (in connection with the AUGMENT 101 study that is already mentioned in the manuscript, is
Menin inhibition with revumenib for NPM1-mutated relapsed or refractory acute myeloid leukemia: the AUGMENT-101 study.
Arellano ML, Thirman MJ, DiPersio JF, Heiblig M, Stein EM, Schuh AC, Zucenka A, De Botton S, Grove CS, Mannis GN, Papayannidis C, Perl AE, Issa GC, Aldoss I, Bajel A, Dickens DS, Kühn MWM, Mantzaris I, Raffoux E, Traer E, Amitai I, Döhner H, Greco C, Kovacsovics TJ, McMahon CM, Montesinos P, Pigneux A, Shami PJ, Stone RM, Wolach O, Harpel JG, Chudnovsky Y, Yu L, Bagley RG, Smith AR, Blachly JS. Blood. 2025 May 7:blood.2025028357. doi: 10.1182/blood.2025028357. Online ahead of print. PMID: 40332046
Author Response
Comment: Figure 1: can be ambiguous. Recommending to label this image as panel B, and generate a panel A that allows readers to follow the intended description. Furthermore, write a few sentences in the legend, to let readers understand the figure. Despite being familiar with the topic, it took me to read over four times to follow the image. It may have been easier if the figure was instead divided into two panels. The current format could have been more suitable, after editing for something like a graphical abstract, which indeed has to be understood without legend; the aim then would be to provoke someone to read the article, rather than to understand before reading.
Response: Thank you for your observation. We have revised the figure to enhance its clarity and ensure it is more easily understandable for the readers.
Comment: Other: based on the email address, the spelling for the name Cristina Papyannidis is wrong and needs to be corrected to Cristina Papayannidis.
Response: Thanks, we correct it
Comment: Examine the possibility of adding recent papers:
Response: Thank you for the suggestion. We will update the bibliography to include some recent references available up to 2025 in the final version of the manuscript.
Round 2
Reviewer 2 Report
Comments and Suggestions for Authors
The publication is reproductive and brings few new insights.
Author Response
Comment: The publication is reproductive and brings few new insights.
response :
We are sorry to read your impression that the manuscript is largely reproductive and offers limited novel insights. We understand your concerns, particularly considering that the topic of menin inhibitors has become increasingly popular and may appear overrepresented in the literature. It is possible that the original aim of our review was not sufficiently clear in its earlier form.
However, we have revised the manuscript extensively according to the valuable guidance provided by the Academic Editor. We hope that the changes implemented — including better contextualization, clarification of our objectives, and a more focused analysis — will enhance the clarity and overall usefulness of the review.